# Assessing the Influence of Social Factors on Local Perceptions of Climate Change, Product Value Addition, Multiple Uses of NTFPs, and Their Influence on Poverty Alleviation in Ghana

**Obed Asamoah** [1,*] 📷, **Jones Abrefa Danquah** [2], **Dastan Bamwesigye** [3] 📷, **Mark Appiah** [4] **and Ari Pappinen** [1]

1 School of Forest Sciences, University of Eastern Finland, Yliopistokatu 7, P.O. Box 111,
 FI-80101 Joensuu, Finland; ari.pappinen@uef.fi
2 Department of Geography and Regional Planning, Faculty of Social Sciences, College of Humanities and
 Legal Studies, University of Cape Coast, Cape Coast 033, Ghana; jones.danquah@ucc.edu.gh
3 Faculty of Forestry and Wood Technology, Mendel University in Brno Zemědělská, 361300 Brno,
 Czech Republic; xbamwesi@mendelu.cz
4 CSIR College of Science and Technology (CCST), Accra P.O. Box M32, Ghana; anmark.appiah@gmail.com
* Correspondence: obeda@uef.fi; Tel.: +420-7775-92119

**Abstract:** The key to the successful and sustainable utilisation of non-timber forest products (NTFPs) by local communities lies in understanding their perspectives on climate change and its effects on NTFP production. Furthermore, assessing their perceptions of the diverse uses and potential for the value addition of NTFPs is crucial for determining how these resources can enhance the well-being of local livelihoods. Although studies on climate change, value addition, and the multiple uses of NTFPs and their impact on local livelihoods have been conducted, locals' perceptions of these factors have not been fully explored. This study aims to assess various social factors, including education, gender, and religion, and their influence on locals' perceptions of the abovementioned factors. The research focused on five regions where NTFPs are prevalent in Ghana (the Western North, Bono, Ahafo, Ashanti, and Eastern regions). To investigate the influence of social factors on local perceptions of climate change, value addition, and the multiple values of NTFPs within rural communities, a total of 732 locals were randomly selected with a team of interviewers across these five regions in Ghana. Key informant interviews and focus group interview methods were used for the data collection. We applied a logistic regression model to assess the social factors and their influencing locals' perceptions of the mentioned variables. It was revealed that age, gender, education, and occupation did not significantly influence locals' perceptions of climate change. However, religion was found to influence locals' perceptions of climate change. Furthermore, the results demonstrated that gender and education strongly influenced locals' perceptions of value addition to NTFPs. In contrast, religion and constraints (related to time, finance, and skills) did not significantly influence locals' perceptions. In conclusion, this study provides valuable insights into the intricate interplay among local perceptions, climate change, value addition, multiple uses of NTFPs, and the capacity of NTFPs to enhance the well-being of forest fringe communities. These findings underscore the importance of comprehensive, tailor-made interventions that consider local perspectives and adhere to sustainable approaches, with the aim of optimising the positive impacts of NTFPs in poverty alleviation and overall community development.

**Keywords:** non-timber forest products; perception; climate change; value addition; multiple use

## 1. Introduction

In countless communities around the world, non-timber forest products (NTFPs) play an essential role in sustaining ecosystems and livelihoods [1–3]. The term NTFPs refers to a wide variety of plant- and animal-based resources harvested from forests, unlike timber, which revolves around the extraction of wood [4–6]. Their importance stems from their

diverse and multifaceted contributions extending far beyond their immediate economic value. Food, medicine, materials, and other essentials that local communities require for subsistence are supplied by NTFPs [7,8]. Culturally significant, NTFPs often represent traditional knowledge and practices that have been passed down from generation to generation [9,10]. NTFPs also have the potential to improve food security by diversifying diets and supplementing incomes, particularly in areas where agriculture may be less reliable [11,12]. Again, biodiversity conservation relies heavily on NTFPs. The sustainable harvesting of NTFPs can help preserve forest ecosystems and their unique acknowledgement of the multiple uses of NTFPs, from medicinal to industrial applications, encourages a comprehensive approach to resource utilisation [13,14]. Furthermore, NTFPs provide a significant source of income for many forest-dependent communities [15,16]. NTFPs can be harvested and sold locally or globally, creating economic opportunities while reducing pressure on timber resources [17,18]. A variety of value-added products can also be made from NTFPs, such as herbal medicines or handicrafts, to increase their economic potential [19,20].

Recognising the significance of climate change, value addition, and the multiple uses of NTFPs is crucial for formulating sustainable policies [21]. Climate change considerations are vital to ensure the conservation of ecosystems and their role in mitigating environmental impacts [22,23]. Value addition promotes economic growth by enhancing the marketability of NTFPs and fostering economic sustainability [12,19]. Integrating these aspects into policy frameworks ensures the long-term viability of NTFPs and supports the balance between environmental conservation and socioeconomic development [24,25].

To establish sustainable policies for the utilisation of NTFPs, it is essential to examine the local perception of climate change, value addition, and multiple uses of NTFPs [26,27] and other social factors that influence their perceptions [16]. Understanding the perspectives of local communities can help identify challenges, opportunities, and potential strategies for promoting sustainable NTFP utilisation [21]. In Ghana, people's perceptions regarding resource utilisation, including NTFPs, are significantly influenced by social factors, for example, education, religion, gender, and tribal affiliations [28,29]. Education can play a crucial role in shaping people's perceptions of resource utilisation [30]. People's perceptions and engagements with resource utilisation are transformed by education [31]. When individuals are equipped with knowledge and insights into the complex environmental and social implications of NTFP extraction, it allows them to make informed decisions and advocate for more sustainable practices within their local communities [32,33]. An integral part of this transformation is the capacity of education to instil a deep understanding of the multifaceted consequences of NTFP extraction [34]. Through education, individuals become attuned to the ecological dynamics, such as the potential impact on forest ecosystems, biodiversity, and overall ecosystem health [35]. In addition, they gain insight into the social dimensions, including the effects on local livelihoods, indigenous knowledge systems, and community dynamics. In the NTFP value chain, education can increase awareness of the benefits of value addition [36,37]. A higher level of education increases locals' understanding of the potential for income generation through value addition [38,39]. In addition to improving access to market information, education can help locals identify value-added opportunities and target specific markets. With this knowledge, they can make informed decisions as to which NTFPs to add value to and how to effectively market their products [40]. A comprehensive education program empowers individuals to engage in dialogue and advocacy, influencing policies and initiatives that support the sustainable utilisation of NTFPs [41].

Religion can have a significant influence on locals' perceptions of climate change. In Ghana, where the impact of climate change is increasingly being felt, religion plays a vital role in shaping people's understanding of and response to environmental issues [42–44]. Ghanaian communities may attribute the changes in the climate and their effects on the environment to God or spiritual forces influenced by their religious beliefs. This perspective can shape their perception of climate change as a natural occurrence beyond human control, potentially affecting their motivation to take action [45]. The concept of stewardship

is emphasised in Ghanaian religious teachings, particularly those of the Christian and Islamic faiths [43]. This perspective can lead to a greater sense of responsibility and motivation to address climate change and sustainable utilisation of NTFPs. Religious leaders can influence the perception of their followers regarding climate change and its influence on other resources in Ghana. Researchers have explored how Christian, Islamic, and indigenous African religious leaders view climate change challenges and how they can address them [43,46]. The messages they convey and their actions can impact local perceptions and motivate them to adopt sustainable practices. Religion can positively and negatively influence locals' perceptions regarding climate change and the sustainable use of NTFPs in Ghana.

The use of NTFPs can have symbolic meaning and use-value in the life of different religious groups. Religious ceremonies, such as marriage, death, and other worship and ceremonial practices, may utilise some NTFPs [47,48]. Some NTFPs may be taboo or have a specific religious significance, affecting how they are used and managed. These taboos and restrictions can influence the availability and utilisation of certain NTFPs in local communities [38]. Religious beliefs and practices can also influence the collection and processing of NTFPs. Studies have shown that religious beliefs significantly impacted the collection of NTFPs, with farmers with different religious beliefs participating in NTFP collection processing and commercialisation [49]. NTFPs can play a role in preserving cultural practices and traditions associated with religious beliefs. For example, religious and cultural practices can influence using bamboo and rattan in crafts, toothbrushes, and other daily items [50].

Local perceptions of climate change can be influenced by gender in many ways. A study conducted in Ghana's Pra River Basin confirmed that locals perceive climate change differently, are more vulnerable, and adopt different adaptation strategies based on their gender [51]. According to one study [51], men were perceived as more vulnerable to temperature and rainfall events, while women were perceived as more vulnerable to droughts. In addition to income, education, and occupation, gender often plays a vital role in determining an individual's vulnerability to climate change [52,53]. Due to their roles in resource-dependent activities, limited access to resources, and restricted mobility, women are disproportionately vulnerable in many societies [54,55]. Consequently, women may be more sensitive to the effects of climate change on their communities. Education and empowerment disparities can affect people's awareness and understanding of climate change [56]. It is possible that women with limited access to education may lack information about climate change and its implications, which may affect their perceptions of the issue. The perception of locals regarding the value addition and commercialisation of NTFPs can be influenced significantly by gender [19,57,58]. Most rural women are heavily involved in the collection and processing of NTFPs, which are often crucial sources of income and livelihood for their families [59,60]. However, several factors can limit women's ability to engage in value-added activities and fully benefit from the commercialisation of NTFPs. For example, it is not uncommon for rural women to have multiple responsibilities, including household chores, childcare, and agricultural work. It can leave them with little time and energy for value-added activities for NTFPs [61,62]. Women may face challenges accessing the necessary resources and skills to add value to NTFPs. For example, they may not have access to processing equipment or training on value-added techniques. Other NTFPs are only meant for men in processing them. For example, Hunting and fishing activities are often associated with men in traditional societies due to their physical demands, such as strength and endurance [63,64]. The men may be responsible for catching fish, hunting game, or trapping animals for their hides. Sometimes, medicinal plants, herbs, and roots are collected and prepared differently by men and women [65,66]. A particular medicinal plant and its preparation method may be known to men in certain cultures. Some NTFPs are used in the creation of crafts or artwork. Men may be involved in crafting items like wooden carvings, while women might focus on other craft activities [67]. Traditional roles and skill sets may influence the division of labour. The act of collecting wild honey from beehives

located in forests or other natural environments is considered a male activity in some regions due to the risk involved in climbing trees and gaining access to beehives [68,69].

Local communities can be significantly impacted by gender in the commercialisation of non-timber forest products (NTFPs). Women and men have similar access to NTFPs in the wild, with restrictions occurring only on plots of land in cultivation or fallow, where the landowner and their close relatives have exclusive access to the products. It is possible, however, for men and women to have different preferences for forest products [70,71]. Most NTFPs are collected and processed by men, while most NTFPs are marketed and sold by women [69]. Due to this division of labour, men may have difficulty accessing markets and obtaining fair prices for their NTFPs. Men tend to control higher-value NTFP resources or may appropriate resources previously managed by women if their commercial value increases. As a result, benefits derived from commercialising NTFPs may be inequitable for women and men [57].

Several studies have been conducted to assess the potential of NTFPs to alleviate poverty [20,38,72] in Ghana and other parts of the world. Again, other studies have also assessed the impact of climate change, value addition, and multiple uses of NTFPs and their potential to improve local livelihoods. However, the assessment of local perceptions and the various influencing social factors (education, gender, religion, and age) about climate change, the value addition of NTFPs, and the diverse uses of NTFPs have been insufficiently addressed [26,73]. The study aims to assess the factors that influence the perception of locals on climate change, value addition, and multiple uses of NTFPs in local communities in Ghana. The study states a question: Do social factors like education, gender, age, and religion influence local perceptions of climate change, value addition, and multiple uses of NTFPs?

## 2. Methodology

### 2.1. Study Area

This study focused on five regions in Ghana, namely, the Western North, Bono, Ahafo, Ashanti, and Eastern regions, where NTFPs hold prominence. These regions are renowned for their diverse range of NTFPs. The primary economic activities in these selected areas revolve around agriculture (farming), trade, commerce, and services, encompassing hotels, auto mechanics, sawmills, and banks. The study areas are characterised by lush and varied vegetation, including towering canopy trees, understorey plants, and a rich array of plant and animal species, as documented by [39] (Figure 1). The study area is in the Tropical Rainforest Climatic Zone, characterised by various tree species such as Onyina, Odum, Wawa, Mahogany, Sapele, Emire Asamfina, Red cedar, and others. The average annual rainfall varies between 1524 and 1780 mm. The region experiences a bimodal rainfall pattern, with the primary season's peak occurring during June–July and the secondary season's peak from September to October. Monthly mean temperatures range from 25 to 30 °C. In the hottest months, notably February and March, temperatures can soar to 31–33 °C, while in the coldest month, August, temperatures can dip to 19–21 °C [39].

### 2.2. Method of Survey and Study Approach

To investigate the local perspectives on climate change, value addition, and the multifaceted values associated with NTFPs in rural communities, a comprehensive study involved 732 interviews. These interviews were conducted by a team of interviewers across five regions in Ghana, specifically the Ahafo, Ashanti, Bono, Eastern, and Western North regions. The initial step in this research process was the development of a preliminary questionnaire in March 2022. This questionnaire with an initial 30 questions was created (closed- and open-ended). After extensive consultations with economists, market stakeholders, community members, and the Ghana Forest Commission representatives, the questions were streamlined to 25. To ensure the questionnaire's accuracy and conceptual clarity, it underwent a meticulous review by three lecturers: one from the School of Forest Science at the University of Eastern Finland, another from the University of Cape Coast in Ghana,

and a third from the Forest Research Institute of Ghana. Their invaluable input played a significant role in refining the questionnaire's language and ensuring its conceptual clarity. Data collection occurred between April and June 2022, adhering to a well-defined schedule. During this period, interviews were conducted in various regions, districts, and local communities. In addition to individual interviews, focus group discussions were organised in June 2022 within the proximity of the study areas. These discussions aimed to gauge the comprehension of NTFP collectors, marketers, and farmers regarding climate change, value addition, and the diverse uses of NTFPs in Ghana, as well as their potential to improve their quality of life. In-person interviews in selected communities within the regions were carried out from May to June 2022. The subsequent phase involved data collection, cleaning, quality checks, and data management, conducted between July and August 2022. This phase received valuable assistance from one of the lecturers from the University of Cape Coast in Ghana.

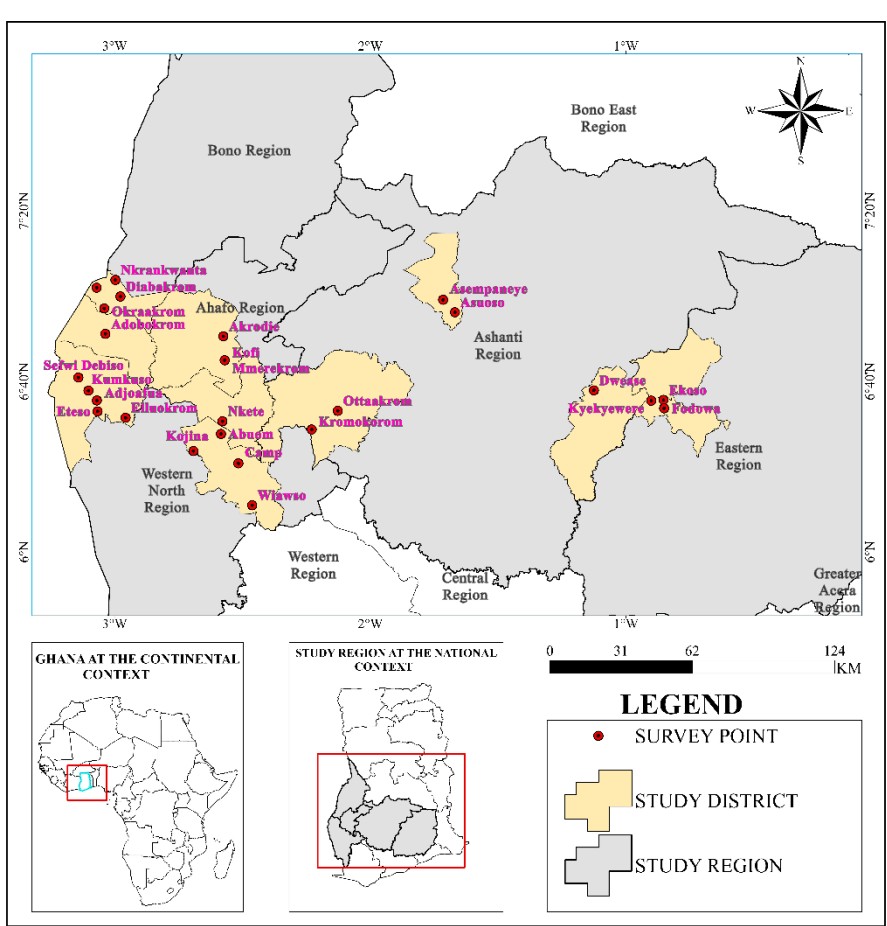

**Figure 1.** The map of study sites showing the regions and selected communities—ref [39].

### 2.3. Sample Design and Data Collection

In the designated regions, the study encompassed residents of districts, communities, and villages adjacent to the forests. The study participants were drawn from households, market areas, and public centres within these district, community, and village settings. Certain criteria were established to identify eligible individuals for the study, which excluded non-Ghanaians, those who had not resided in the district for at least two years, and children under the age of 18. The selection of districts within the study area was based on their proximity to forest reserves. In contrast, the choice of communities within these districts depended on the number of reserves in the district and their nearness to the forest, whether within or outside the reserve boundaries. A purposeful sampling method was employed to identify and select respondents (732). This approach was employed

to ensure the collection of in-depth insights from local individuals actively engaged in the utilisation of non-timber forest products (NTFPs). It involved the identification and selection of individuals or groups possessing substantial knowledge or experience concerning the specific phenomenon of interest, following the principles described by [74]. Additionally, key informant interviews and focus group discussions were conducted with selected participants, including hunters, traders, and processors, who directly engaged in the utilisation of NTFPs about the impact of climate change and its influence on NTFP production. The purposeful sampling method was instrumental in ensuring that every member of the selected population had valuable insights related to NTFPs, thus minimising the likelihood of inconsistent responses and upholding the sample's representativeness concerning accurate information. Cochran's formulas and procedures were applied to determine the appropriate sample size.

To calculate the sample size, we used [75] formulas and procedures as described in [76]. The overall sample size (Nfs) for this study, which was 732 individuals, was stratified and considered various demographic and contextual factors. These factors included age, gender, educational background, religion, occupation, the duration of respondents' residence in the community, the specific types of non-timber forest products (NTFPs) they harvested, and any changes in the NTFPs they collected. Among the NTFPs investigated in the study were mushrooms, snails, chewing sticks, games, and honey. The selection of these products was based on their anticipated significance in the NTFP trade within the study areas.

### 2.4. Data Analysis

The primary objective of this study was to explore how local communities perceive climate change, value addition, and the diverse applications of non-timber forest products (NTFPs). To achieve this goal, we categorised interviewees based on their roles within the NTFP value chain, namely, collectors, buyers, and consumers. For the analysis of descriptive statistics in the study, we utilised SPSS Statistics 20.0 (IBM, New York, NY, USA) and R Studio. We evaluated the survey results, focusing on the frequencies and percentages of responses related to each indicator pertaining to the locals' viewpoints on climate change, value addition, and the multiple uses of NTFPs. Additionally, data gathered from household interviews underwent analysis using SPSS and R Studio to compute descriptive statistics, such as percentages and frequencies. The findings were then presented in tables and figures. We employed logistic regression analysis to ascertain the significance of local people's perceptions regarding climate change, value addition, multiple NTFP uses, and their connection to socioeconomic factors. In particular, we applied a logistic regression model to assess the factors influencing farmers' perceptions among the locals and their potential to enhance their livelihoods. We selected an ordinal logistic regression model, a commonly used approach for studying the factors influencing farmers' views on various technologies, as established in previous research [77,78]. The logistic regression method was employed since the dependent variable was binary, represented as 0/1, True/False, or Yes/No. Within a binomial distribution, the logit function was the link function, following established statistical practices [79–81].

### 2.5. Analytical and Conceptual Framework

Methods for analysing the local perceptions of climate change, value addition, and the multiple uses of NTFPs often align with the extensive literature on the adoption [82,83] of NTFP production technologies, with a predominant focus on new or improved production inputs. Most studies on the adoption of NTFPs have employed logit or probit models to analyse binary adoption decisions, where the dependent variable is dichotomous [84,85]. The modelling of local perceptions regarding climate change, value addition, and the multiple uses of NTFPs can be accomplished by applying a dichotomous or binary-dependent logit model as described by [86,87]. Furthermore, the perception of locals on value addition, climate change, and the multiple uses of NTFPs can be shaped by various independent biophysical and socioeconomic factors [87,88]. The logit regression statistical analysis tech-

nique used in this study is explained in several studies, for example, [89–92]. For simplicity, let us denote the perception of locals as 'Y', where Y equals 1 for a positive response and 0 otherwise. The foundational comprehension function, assessing the understanding of locals, is presumed to be a function of locale-specific attributes (represented by the vector X, encompassing socioeconomic features of locals and village-specific characteristics). The utility function for perceptions of locals on utilisation of NTFPs considering (climate change, value addition, and multiple uses) is $Ui_l = \beta_i X_i + \varepsilon_{il}$, and the function for non-participation is given as $Ui_0 = \beta_0 X_I + \varepsilon_{il}$, where $\varepsilon$ is an error term with a logistic distribution. Because the utilities are random, the *i*th local will select the alternative 'adoption' if and only if $Ui_l > Ui_0$.

Thus, for local *i*, the probability of perceptions of the locals on NTFP utilisation 't' is given by the following:

$P((Y_{it} = 1) = P(U_{il} > (U_{i0})$ and the methods as proposed by [91].

Locals, both males and females in the forest communities, are responsible for collecting the non-timber forest products (NTFPs) depicted in Figure 2. Notably, certain NTFPs like leaves, snails, and mushrooms are predominantly gathered by women due to their accessibility and less strenuous collection, processing, and sales activities, confirming findings from previous research [93]. Specifically, 50 women, constituting 6.83%, engage in the comprehensive process of mushroom gathering, utilisation, processing, and sales, driven by the prevalent use of mushrooms in home-cooked meals. Additionally, 65 women, equivalent to 8.88%, are actively involved in collecting, processing, and selling leaves, often used for wrapping food sold along roadsides and in modern restaurants to reduce plastic pollution. In contrast, only 20 men, representing 2.87%, explicitly mentioned their participation and sales of NTFPs. Notably, snails emerge as the most significant NTFP for women, with 120 individuals (16.39%) engaging in their collection, processing, and sales. The study reveals that snails are highly sought after in the market, and women invest substantial time in gathering and processing. While some snails are sold directly from the forest without processing, the processed ones command higher prices and are often utilised for household meals. Conversely, certain NTFPs such as game, herbs, Raffia Palm, straw, honey, and pestles were predominantly managed by men. For instance, game hunting involved 50 male respondents, constituting 6.83%, while no women reported participating in game hunting (0%). This disparity may be attributed to the strenuous and often nocturnal nature of game hunting. Raffia palm, essential for crafting baskets and mats, witnessed significant male involvement due to the labour-intensive process in its preparation for drying products like cocoa. Honey hunting and collection, though deemed time-consuming, were acknowledged as valuable and lucrative activities.

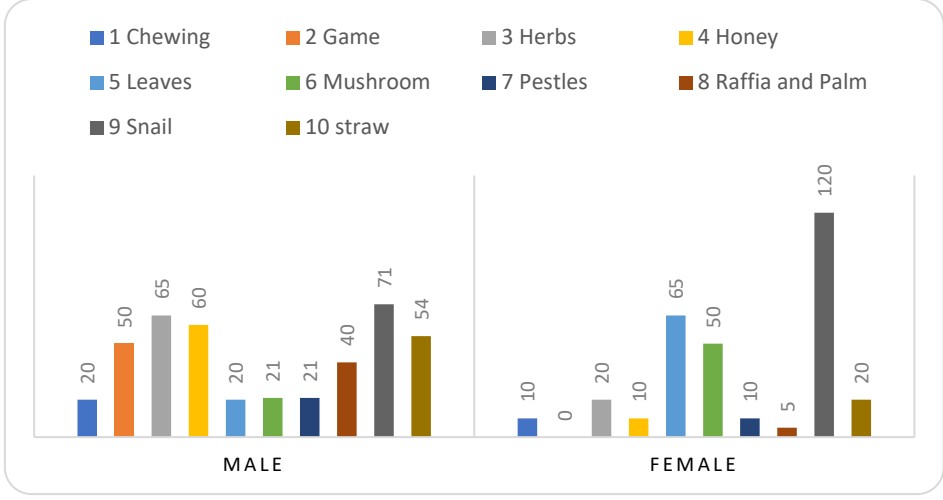

**Figure 2.** The collection of NTFPs in the study area by males and females.

## 3. Results

Due to the traditional family structure, where men are typically regarded as the household heads responsible for gathering and hunting non-timber forest products (NTFPs), a more significant proportion of male participants were included in this study. Furthermore, it was observed that many women exhibited a more reserved attitude when responding to questions, frequently seeking approval from their husbands before actively engaging in the survey.

A survey was carried out, encompassing a total of 215 female respondents (29.37%) and 517 male respondents (70.63%), as detailed in Table 1. Most individuals surveyed fell within the age group of 50 to 59, accounting for 298 respondents (40.71%). In contrast, the age category 18–20 had the lowest representation, with just 5 respondents (0.68%). This lower participation can be attributed to the timing of the survey, which coincided with school hours, and the fact that many individuals in the 18–20 age group were likely attending school during that time. The education levels within the local population exhibited variability, with a significant portion having completed only primary education. Among the survey respondents, 371 individuals (50.68%) had attained only primary education, while 41 (5.60%) held graduate degrees. The educational distribution may be influenced by financial constraints prevalent in the study area. Regarding the religious beliefs of the surveyed residents, 438 individuals, accounting for 59.8%, identified as Christians. In contrast, 35 respondents (4.8%) identified as Muslims, and 259 individuals were recorded as adherents of traditional beliefs. In the survey area, various occupations were reported, with farming being the most prevalent, involving 620 respondents, accounting for 88.4% of the participants. Trading was the second most common occupation, with 109 respondents representing 11.2%. Teaching and nursing, on the other hand, were the least common, with only 2 (0.3%) and 1 (0.1%) individuals engaged in these professions, respectively.

From the responses of the locals from the survey with regards to their perception of climate change, it was observed that 208 of the respondents, representing 28.4%, mentioned that human activities have caused the changes of the climate, 190 of the respondents, representing 25.9%, specifically attributed the changes in the climate to the rampant illegal logging of trees (Table 2), and 75 respondents, representing 10.3%, mentioned that industrialisation has contributed to the changes in the areas. Even though some locals mentioned anthropogenic causes of climate change, others attributed it to spiritualities. A total of 54 of the respondents mentioned that climate change is the result of sins that humans cause and 51 of the respondents claimed that it is due to the bad behaviour of humans, i.e., the gods of the land are angry with mankind, which has resulted in drastic changes in the weather patterns and changes in the climatic conditions. Among all the people contacted, 52, representing 7.2%, made it known to the interviewers that they are uncertain about the changes in the climate, and most of these changes are bound to occur and are natural phenomena.

In discussions with the local forest fringe communities about the impacts of climate change on their lives, their perspectives were organised into categories based on perceived changes, the consequences of their daily existence, and the specific transformations they had observed (Table 3). Locals mentioned perceiving erratic, delayed, and untimely rainfall patterns. They also observed decreased water availability and increased dry conditions. Furthermore, they noted rising temperatures and shorter, warmer rainy seasons, resulting in reduced rainfall, which, in turn, led to an increase in pests and diseases.

**Table 1.** Sample statistics of the explanatory variables (social factors) used in the econometric model and their hypothesised signs. Values in parentheses indicate frequencies or proportions.

| Variable | Description | Frequencies | Continuous Variable | | Categorical Variable (%) |
|---|---|---|---|---|---|
| | | | **Mean** | **Std. Dev** | |
| Age | Age of the locals in years. Ages considered were between 18 and 60 | | | | |
| | 18–20 | 5 | 47.88 | | |
| | 21–29 | 115 | 14.19 | | |
| | 30–39 | 89 | | | |
| | 40–49 | 95 | | | |
| | 50–59 | 298 | | | |
| | 60 and -older | 130 | | | |
| Gender | Gender of the locals<br>1 = male, 0 = female | 0 = 215<br>1 = 517 | | | 1 = 29.2<br>0 = 70.5 |
| Education | Locals' level of education category, High school = 1, Primary school only = 2, High school no degree = 3, Some Graduate Level Courses = 4 | 1 = 72<br>2 = 371<br>3 = 248<br>4 = 41 | | | 1 = 9.8<br>2 = 50.7<br>3 = 33.9<br>4 = 5.6 |
| Religion | Religion encompasses belief systems, values, and practices associated with sacred or spiritual matters, influencing the way individuals interact with the natural environment. In Ghana, approximately 71% of the population adheres to Christianity, 18% to Islam, and a smaller percentage follows traditional beliefs [86]. For coding purposes, Christian affiliation is represented as 1, while Traditional affiliation is coded as 2 and Muslims as 3. | 1 = 438<br>2 = 259<br>3 = 35 | | | 1 = 59.8<br>2 = 35.4<br>3 = 4.8 |
| Occupation | Describes the sector of the economy where locals are gainfully employed.<br>For coding purposes, farmers are represented as 1, Teachers as 2, Nurses as 3, and Traders as 4. | 1 = 620<br>2 = 1<br>3 = 2<br>4 = 109 | | | 1 = 88.4<br>2 = 0.1<br>3 = 0.3<br>4 = 11.2 |
| Use of NTFPs by locals | The describes the multiple uses of NTFPs by locals. This is where the utilisation of NTFPs was categorised and coded as follows.<br>Artefacts = 1 Construction = 2 Food = 3, leisure = 4, Medicine = 5, Religion = 6, and sale = 7 | 1 = 112<br>2 = 95<br>3 = 180<br>4 = 15<br>5 = 115<br>6 = 45<br>7 = 170 | | | 1 = 15.3<br>2 = 13<br>3 = 24.6<br>4 = 2<br>5 = 15.7<br>6 = 6.1<br>7 = 23.2 |

**Table 2.** Locals' perceptions of the causes of climate change.

| Causes | Response (n) | Percentage |
|---|---|---|
| Human activities | 208 | 28.4 |
| Punishment from gods | 51 | 6.9 |
| Sinful act | 54 | 7.4 |
| Population increment | 102 | 13.9 |
| Industrialisation | 75 | 10.3 |
| Uncertain | 52 | 7.2 |
| Tree cutting | 190 | 25.9 |
| Total | 732 | 100 |

The local communities openly shared the adverse impacts they had experienced. They expressed concern about the dwindling availability of non-timber forest products (NTFPs), which had affected their ability to collect them. They also revealed that these climate-induced changes had repercussions for NTFPs and agricultural production. They

mentioned that shortened and unpredictable rainfall seasons had negatively affected soil moisture conditions, leading to decreased groundwater and impacting plant growth, which, in turn, affected other organisms that contribute to NTFP production.

**Table 3.** Locals' responses to climate and socioeconomic change.

| Perceived Change | Impact Experienced on the Livelihood System | Response (Observed Changes) | Study Area |
|---|---|---|---|
| Erratic, untimely, delayed, and rainfall | The decline in NTFP collections and gathering and agricultural productivity. Decline in NTFP production, e.g., honey, snails, and mushroom producers | Changes in NTFP collection season Delayed or earlier sowing and harvesting Changes to crop varieties and types Labour migration Increased engagement in wage labour Re-sowing of crops Buying food from the market Skipping meals Sale of assets Borrowing money | All regions and districts |
| Overall, decreased water availability and increased dry conditions. | Less flow of rivers, springs and streams. Drying up of moisture; lowering of groundwater or soil moisture and forest. | Scarcity of water for NTFP processing and NTFP production Scarcity for NTFP collection and gathering. Rampant recording of forest fire Less land area cultivated | All regions and districts |
| Increasing temperatures | Health issues (increased incidence of heat stroke and vector-borne diseases). | Increased expenditure in treatment of ailments. Increased crop failure. Increased livestock failure. Increased NTFP failure in the season affecting the financial state of locals. | All regions and districts |
| Warmer and shorter rainy seasons with less rainfall Snowfall/Increase in pests and diseases | Beneficial conditions for specific crops and consequential loss of farm AND NTFP income Increased incidence of pests and disease | Failure of NTFP season for collection and gathering Traditional pest management strategies (Burning fields; spreading cow urine, salt, and ashes; crop rotation) Increased use of pesticides | All regions and districts |

Additionally, the locals mentioned that the rising temperatures could result in health issues for plants and animals, further impacting NTFP production and sales. These changes not only disrupted NTFP production but also increased expenses related to pest and disease treatment, raising household costs. In some cases, the increased temperatures could lead to crop and livestock failures, posing a significant threat to the local community's livelihood. According to the local community members, climate change has altered the timing of NTFP collection. These changes have forced them to seek wage labour, purchase food from the market, skip meals, sell assets, and even borrow money to cope with the new challenges posed by climate change.

The study used logistic regression to examine how various social factors, such as gender, education, religion, occupation, and age, influence the locals' perceptions of climate change and its impact on non-timber forest products (NTFPs, as shown in Table 4). The results indicated that age, gender, education, and occupation did not significantly affect how locals perceived climate change and its influence on NTFPs. However, religion emerged as a factor influencing the locals' perceptions of climate variability, with a statistically significant variability ($p < 0.037$ *). This observation may be attributed to the solid religious orientation of most locals, as religious beliefs can shape one's understanding of environmental changes. Education did not appear to exert a significant influence, possibly because changes in weather patterns are readily observable even without formal education.

**Table 4.** A logistic regression model of people's perceptions of climate change and variability.

| Variable | Estimate | Std Error | z Value | Pr (>\|z\|) |
|---|---|---|---|---|
| Intercept | 18.25583 | 1864.33824 | 0.010 | 0.992 |
| Age | 0.02890 | 0.02546 | 1.135 | 0.256 |
| Gender | −0.11723 | 0.76231 | −0.154 | 0.878 |
| Education | 0.24554 | 0.51858 | 0.473 | 0.636 |
| Religion | −0.99760 | 0.47821 | −2.086 | 0.037 * |
| Occupation | −13.86112 | 1864.33710 | −0.007 | 0.994 |

* = Significant at $p < 0.05$.

In a survey of 732 respondents, it was found that 74.9% of locals do not engage in adding value to NTFPs. However, 59.7% of respondents believed that adding value to NTFPs could increase their sales and enhance their commercialisation within the local community instead of selling them directly. Religious beliefs did not seem to influence their views on NTFP value addition, with 59.8% identifying as Christians, 4.8% as Muslims, and 35.4% as Traditionalists (Table 5). Regarding constraints, 23.6% of respondents cited financial limitations as barriers to waiting for value addition. Some NTFPs, such as mushrooms, snails, and leaves, were mentioned as having lower prices after value addition. Additionally, 39.9% expressed a need for skill development in NTFP value addition. Lastly, 36.2% preferred selling NTFPs immediately after collection due to time constraints, as some products required significant processing time and attention, making direct sales a more practical option.

**Table 5.** Responses of the locals regarding their perceptions of value addition.

| Response | Freq | Percent | Valid Percent | Cumulative Percent |
|---|---|---|---|---|
| Value Addition | | | | |
| No | 548 | 74.9 | 74.9 | 9.8 |
| Yes | 184 | 25.1 | 25.1 | 100.0 |
| Total | 732 | 100.0 | 100.0 | |
| Increased Price | | | | |
| Yes | 437 | 59.7 | 59.7 | 59.7 |
| No | 295 | 40.3 | 40.3 | 100.0 |
| Total | 732 | 100.0 | 100 | |
| Religion | | | | |
| Christians | 438 | 59.8 | 59.8 | 59.8 |
| Muslims | 35 | 4.8 | 4.8 | 64.6 |
| Traditional | 259 | 35.4 | 35.4 | 100.0 |
| Total | 732 | 100 | 100 | |
| Constraints | | | | |
| Financial | 175 | 23.9 | 23.9 | 23.9 |
| Skills | 292 | 39.9 | 39.9 | 39.9 |
| Time | 265 | 36.2 | 36.2 | 100.0 |
| Total | 732 | 100.0 | 100.0 | |

The study examined the impact of gender, education, and various constraints (time, skills, and finance) on local perceptions regarding value addition to NTFPs, as shown in Table 6. The results, obtained through logistic regression, revealed significant influences: gender and education strongly influenced the locals' perceptions of NTFP value addition, as indicated by *p*-values of 0.000328 and 0.00426, respectively. However, the study did not find any influence of religion or constraints (time, finance, and skills) on the locals' perceptions.

**Table 6.** A logistic regression model of people's perceptions of value addition and what issues related to it.

| Variable | Estimate | Std. Error | z Value | Pr (>\|z\|) |
|---|---|---|---|---|
| (Intercept) | 1.1129138 | 0.4782433 | 2.327087 | 0.0199 * |
| Gender | −0.6493546 | 0.1807774 | −3.592012 | 0.000328 *** |
| Education | 0.3507588 | 0.1227342 | 2.857873 | 0.004265 ** |
| Constraints | 0.1706921 | 0.1127015 | 1.514550 | 0.1298864226 |
| Religion | −0.2244204 | 0.1468319 | −1.528418 | 0.1264088668 |

*, **, *** = Significant at $p < 0.05$, $p < 0.01$, $p < 0.001$, respectively.

*Response of Locals on Perception of Multiple Values of NTFPs and Their Potential to Alleviate Poverty*

Locals' perceptions of NTFP uses were categorised into several purposes: artefacts, construction, food, leisure, medicine, religion, and sales. Among the respondents, 15.30% mentioned collecting NTFPs for making artefacts for sale, while 24.59% perceived NTFPs primarily for food use. A smaller percentage, 2.05%, believed NTFPs could be used for leisure (Figure 3). Some locals also considered NTFPs for religious purposes, and a significant portion mentioned NTFPs for medicinal applications (15.71%). Additionally, 23.22% of respondents perceived NTFPs as products for sale, and 12.98% reported using NTFPs for construction purposes.

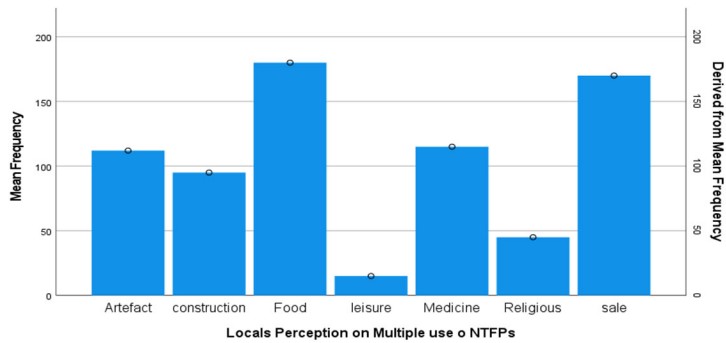

**Figure 3.** Locals' perceptions of the multiple uses of NTFPs by the locals in the study area.

The study employed logistic regression to analyse the impact of various factors on local perceptions regarding the multiple uses of NTFPs. The factors assessed included gender, education, religion, the multiple uses of NTFPs, and household size. The results indicated that gender, education, and religion significantly influenced how locals perceive and use NTFPs, with *p*-values of 0.00132 **, 0.00692 **, and $8.12 \times 10^{-5}$ *** (Table 7). Gender played a role in the harvesting, gathering, and selling of NTFPs, influencing how NTFPs were perceived. It was observed that household size and the various uses of NTFPs had limited influence on locals' understanding of the diverse applications of NTFPs. Additionally, the nature of work within local communities did not seem to impact their perceptions of NTFP use in these communities.

**Table 7.** A logistic regression model of peoples' perceptions of the multiple uses of NTFPs.

| Variable | Estimate | Std Error | z Value | p Value |
|---|---|---|---|---|
| Intercept | 22.85998 | 1078.87784 | 0.021 | 0.98310 |
| Gender | −1.42808 | 0.44464 | −3.212 | 0.00132 ** |
| Education | −0.82063 | 0.30384 | −2.701 | 0.00692 ** |
| Religion | −1.25122 | 0.31750 | −3.941 | $8.12 \times 10^{-5}$ *** |
| Household size | 0.07156 | 0.07007 | 1.021 | 0.30713 |
| Occupation | −14.09820 | 1078.87703 | −0.013 | 0.98957 |
| Multiple uses of NTFPs | 0.13597 | 0.12301 | 1.105 | 0.26900 |

**, *** = Significant at $p < 0.01$, $p < 0.001$, respectively.

## 4. Discussion

Understanding the causes of climate change is critical for comprehending how local communities perceive its impact on NTFPs [94,95]. Climate change awareness and understanding play a significant role in shaping local perspectives on NTFP availability, quality, harvesting seasons, preservation, and commercialisation changes [96,97]. These factors influence whether locals are willing and able to adapt their traditional NTFP harvesting methods or explore alternative sources. The study revealed that locals have varied perceptions about the causes of climate change, which is in line with the previous studies [98,99], and locals know about climate change and its impact on their environment. Some attribute it to sinful acts and divine punishment, viewing it as a consequence of human behaviour and moral choices [98,99]. Traditional beliefs and superstitions also influence their views, as they believe climate change results from divine punishment or supernatural forces. However, other groups of locals perceive the causes of climate change as a result of industrialisation and illegal logging, and this is not far from the explanation [100,101]. Climate change has severe consequences due to industrialisation and illegal logging, resulting in increased global warming, changes in rainfall patterns, extreme weather events, and disruption of ecosystems. These adverse effects impact NTFP production levels and the livelihoods of local communities.

Climate change leads to shifts in temperature and precipitation patterns, negatively affecting NTFP production, tree mortality rates, and water resources. Climate-induced changes disrupt the ecological balance, with long-term consequences. The study also assessed factors influencing locals' perceptions of climate change, such as education, gender, religion, and occupation. Education, occupation, and gender were found to not influence locals' perceptions of climate change and impacts on NTFPs, contradicting some previous findings [47,102,103]. However, the study conducted by the authors of [45] in the Offinso Municipality of Ghana found that the level of education does not influence the perception of climate change. The direct observation of changes in weather patterns, seasons, and NTFP availability by most locals can shape their perceptions of climate change. Religion was another significant factor, as indigenous African religions, Christianity, and Islam coexist in Ghana. These religions often connect with the environment and natural resources in their beliefs and practices. Religious beliefs and practices can effectively address climate change challenges and influence how local communities perceive and respond to their impact on NTFPs in Ghanaian forest-dependent communities. From the study, it was observed that religion significantly influences ($p < 0.037$ *) the locals' perceptions of climate change and how it affects the locals. This affirms the study conducted by the authors of [104] in three rural South African communities, which found that religious beliefs significantly influence a community's understanding and experience of climate change adaptation.

The local perceptions of adding value to NTFPs play a vital role in their commercialisation and poverty alleviation. Value addition is seen to enhance market opportunities and increase the income generated from these products. Locals strongly believe that adding value to NTFPs will lead to higher commercialisation and increased prices, ultimately improving their livelihoods and reducing poverty. However, the study found that locals tend to add value to their products after sales due to religious beliefs, financial constraints, the time required, and the necessary skills for value addition. The study revealed that religious beliefs and time constraints did not influence (P Values of 0.1264 and 0.1298, respectively) the perception and practices of locals regarding value addition. This contradicts other studies [49]. A study conducted in the Chinese Giant Panda Reserves found that religious beliefs had a significant effect on farmers' collection and processing behaviours regarding NTFPs. Education and gender also play a significant role in shaping how locals perceive the value addition of NTFPs. Education can impact gender roles and attitudes, affecting the local perception of value addition. Gender norms and traditional roles can result in differences in access to NTFPs, with some resources designated for men and others for women based on cultural and physical demands. Research [105–107] has shown that education can play a significant role in shaping gender norms and empowering women in

value chains. Contrary to expectations, constraints related to time, finance, and technology did not significantly influence the locals' perceptions of value addition. The study observed that, due to low incomes, locals may prioritise quick sales of NTFPs to address immediate financial needs rather than investing more time and resources in value addition. Value addition is a critical factor in enhancing the commercialisation of NTFPs and reducing poverty [19]. It can increase the marketability and profitability of these products through processing, packaging, quality control, and diversification. Value addition also empowers local communities, contributes to sustainable livelihoods, and provides economic opportunities related to NTFPs [38]. By commercialising value-added NTFPs, marginalised communities can achieve economic independence and long-term sustainable development, ultimately improving their lives and reducing poverty.

Local communities have a comprehensive understanding of NTFPs, encompassing their various uses, economic value, and potential to alleviate poverty [48]. They are well-acquainted with the different types of NTFPs available in their region and recognise their versatility in serving multiple purposes, including medicine, food, leisure, religion, sale, construction, and handicrafts [38,39]. Moreover, locals know the economic value associated with NTFPs, acknowledging their capacity to generate income and improve household livelihoods. These findings affirm the existing research on the significance of NTFPs in local communities [15,108]. From the study, it was observed that gender, education and religion have a significant influence on the perceptions of the locals on the multiple uses of NTFPs, with their *p* values (Table 7), which is in line with previous studies [36,109]. Gender, education, and religion can have significant influences on the perceptions of locals regarding the multiple uses and value addition of non-timber forest products (NTFPs) in local communities. These findings could be shared with key stakeholders such as religious bodies, the Ghana Education Service, Ghana National Civic Education, the Ghana Forestry Commission, local chiefs, and opinion leaders. This information can contribute to the development and implementation of policies aimed at the sustainable management and utilisation of non-timber forest products (NTFPs).

## 5. Conclusions

This study extensively examined NTFPs and their potential to uplift the livelihoods of forest fringe communities and alleviate poverty. Several key dimensions of local perceptions were investigated, including how locals of different ages, gender, educational, religious backgrounds, etc., perceive climate change's impact on NTFP production, the role of value addition in enhancing NTFP commercialisation and income, and the multiple uses of NTFPs and their potential to improve the well-being of the local community. The study revealed that locals are aware of the effects of climate change on NTFP production, emphasising the importance of understanding and addressing climate change for the conservation and quality of NTFPs. Additionally, the study explored how locals of different social classes perceive the value addition to NTFPs, with findings indicating that some community members refrain from value addition due to time, skills, and financial constraints, hindering their ability to process these forest products. However, they recognise increased income and commercialisation potential if value addition is implemented.

Furthermore, the study shed light on the diverse roles NTFPs play within forest fringe communities. These products are employed for various purposes, including crafting artefacts, construction, food, leisure, medicinal uses, religious rituals, and commercial sales. NTFPs serve subsistence and commercial functions, highlighting their adaptability and potential to enhance the community's well-being. In conclusion, the study provides valuable insights into the intricate interplay of local perceptions, climate change, value addition, and the multiple uses of NTFPs in the context of forest fringe communities. These findings underscore the need for tailored interventions that consider local viewpoints and adhere to sustainable approaches to maximise the positive impacts of NTFPs in poverty alleviation and community development. The study will provide Advancement of Knowledge by contributing to the existing knowledge on sustainable management and utilisation of NTFPs.

Again, the study's findings may provide suggestions for policymakers, offering guidance for the development or adjustment of pertinent policies that consider the local community's perceptions of NTFPs to ensure their sustainable management and utilisation. A significant challenge in our study was the hesitancy of local residents, particularly women, to openly and confidently respond to survey questions. This was driven by concerns (fear and respect) about their husbands. Overcoming this constraint, future research might explore strategies to empower women in the sustainable utilisation of NTFPs.

**Author Contributions:** Conceptualization, O.A. and A.P.; Methodology, O.A. and J.A.D.; Software, O.A.; Validation, O.A. and J.A.D.; Formal analysis, O.A.; Investigation, O.A.; Resources, O.A. and A.P.; Data curation, O.A.; Writing—original draft, O.A.; Writing—review & editing, O.A., J.A.D., D.B. and M.A.; Visualization, J.A.D. and D.B.; Supervision, J.A.D., M.A. and A.P.; Project administration, O.A. All authors have read and agreed to the published version of the manuscript.

**Funding:** This research received no external funding.

**Data Availability Statement:** The Data Statement was already added to the first section when the article was submitted.

**Conflicts of Interest:** The authors declare no conflict of interest.

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
