# Peer review of "Assessing the Influence of Social Factors on Local Perceptions of Climate Change, Product Value Addition, Multiple Uses of NTFPs, and Their Influence on Poverty Alleviation in Ghana"

_forests, doi:10.3390/f15020248_

Round 1
Reviewer 1 Report
Comments and Suggestions for Authors
Abstract is too long; not clear why is important to understand the perception of locals to climate change, value addition, and the multiple uses of NTFPs; it is not clear how interviews were conducted and how data is analysed.
Introduction: fails to explain why the three factors - climate change, value addition, and multiple uses of NTFPs are important in establishing sustainable policies for the utilisation of NTFPs; Are there any factors that should be understand when establishing sustainable policies for NTFPs use?
Methodology:
- the focus groups methodology is not presented in the context of the study;
- data about survey’s strata is missing – number of respondents according to age, gender, educational background, religion, occupation, the duration of respondents' residence in the community, the specific types of non-timber forest products;
- the survey is not properly described – number of questions; types of questions.
- Variables coding from table 1 can potentially reveal some inconsistent and hence biased results – eg only two respondents in nurses occupational code. Why not working with code 1 – farmers; code 2 other occupations. Not clear what means code 4 – traders?
Results:
- Difficult to understand the results from table 2 and 3 without a properly description of the questions used in the study (methodology section);
Discussions fails to reveal how the main findings of the study can be used in future policies designed for the sustainable development of the region.
Author Response
I will, first of all, thank you for the comments and directions, which have helped to put the paper in good shape. Comments are addressed.
Thank you

Reviewer 2 Report
Comments and Suggestions for Authors
The manuscript “Factors that Influence Local Perceptions of Climate Change, Product Value Addition, Multiple Uses of NTFPs, and Their Impact on Poverty Alleviation in Ghana” concerns the important evidence that the climate is undergoing change, and understanding the perception of locals of the magnitude of climate change and its influence on non-timber forest products (NTFPs) challenge at a global scale for Ghana, and will be interested for “Forests’ “ journal readers after revision.
Title: Factors that Influence – it isn’t any factors, only statistical relationships; must be improve;
Method: very simple methods like the Logistic regression model were used;
How the research sample was selected, and how many people participated in five investigated regions? – the lack of this information;
What does “Product Value Addition” mean in this manuscript?
What are the methodical differences between “Product Value Addition” and Multiple Use, and NTFPs?
Table 7: what does “ Intercept “ mean? Unclear; please explain this;
The conclusion is very general and could be drawn without any research in Ghana, almost only “that locals are aware”, without any details (age, gender, religion et al.).
Some specific comments:
492: (M. Li et al., 2020). A study conducted in (…) – be careful with literature only name;
Author Response

(The authors gave the same response as above.)

Reviewer 3 Report
Comments and Suggestions for Authors
Dear Author(s),
Thank you for the opportunity to read the paper entitled Factors that Influence Local Perceptions of Climate Change, Product Value Addition, Multiple Uses of NTFPs, and Their Impact on Poverty Alleviation in Ghana.
The study aimed to investigate factors that influence local perceptions of climate change, product value addition, and the use of non-timber forest products and their impact on poverty alleviation. This topic has not been studied enough, so this study is more than necessary and confirms the interest and originality of this research. It fills the literature gap with a focus on various social factors, including education, gender, and religion, and their influence on locals' perceptions of the abovementioned factors. The study uses a logistic regression model to assess the social factors and their influence on the locals’ perceptions of the mentioned variables.
The manuscript is well-written, understandable, and easy to comprehend. The results are nicely presented through tables and figures. The quality of the presentation is very high. The modification is needed for Figure 2. All cited references are relevant to the research and up-to-date. However, there are references missing in the case study area and this has to be corrected. The conclusion needs to be expanded. What are the theoretical contributions and practical implications of your study? What are the limitations? Future research? These segments are missing. Point out the originality and novelty of your study.
The topic of this paper is interesting, but certain improvements would be appreciated.
Abstract
Comment 1
Having the values of p in the abstract is not necessary. Please remove it. The abstract should make a nice opening to the article to lure the reader and awaken their interest. Values are not needed here.
Introduction
Comment 2
The introduction is very nicely written. There are a variety of references that are relevant to the research. However, it is missing to point out similar research on the topic and what makes your study different. You should point out the research gap as well as the contribution, novelty, and originality of the study. This part should be added at the end of the introduction. You should also include the research questions. So this “aims of the study” heading should be merged at the end of the Introduction and research questions added.
Methodology
Comment 3
Lines 199-206 have no references. Please add them.
Comment 4
The methodology is clear even to someone who is not in this field. Just as it should be, it is excellently explained and described. Great job, author(s)!
Results
Comment 5
The results are nicely presented through tables and figures. Great job! But figure 2 should be modified; some words start with a capital letter, some do not, so adjust it.
Discussion
Comment 6
Line 451 - The study revealed that locals have varied perceptions about the 451 causes of climate change; this confirms the studies of (Buys et al., 2014; Dhungana et al., 452 2018). – Please modify the sentence. You cannot finish it with word “of” and start with “()”. You can say “which is in line with previous research (Buys….).
Comment 7
You do not need to repeat the p values in the Discussion. That is not necessary.
Comment 8
Line 524 – Modify the sentence (same as comment 6).
Conclusion
Comment 9
The conclusion needs rework. What are the theoretical contributions and practical implications of your study? What are the limitations? Future research? These segments are missing. Point out the originality and novelty of your study.
Comment 10
Once again, thank you very much for the opportunity to read this interesting article. The article has serious potential, and it deals with a very popular topic in research, but certain improvements are very much needed. I look forward to reading your article again.
I wish you all the best!
Reviewer
Author Response

(The authors gave the same response as above.)

Round 2
Reviewer 2 Report
Comments and Suggestions for Authors
After revision, the authors suggested the title as follows “Assessing the influence of social factors that Influence Local Perceptions of Climate Change, Product Value Addition, Multiple Uses of NTFPs, and Their influence on Poverty Alleviation in Ghana", but is not changed in the manuscript.
Author Response
The title has been modified.
